# Regionalizing Streamflow Regime Function through Integrations of Geographical Controls in Mountainous Basins

**Shuang Yang** [1,2], **Mengzhu Gao** [3], **Jintao Liu** [1,2,*] , **Pengfei Wu** [1,2] and **Yaqian Yang** [4]

1 State Key Laboratory of Hydrology-Water Resources and Hydraulic Engineering, Hohai University, Nanjing 210098, China
2 College of Hydrology and Water Resources, Hohai University, Nanjing 210098, China
3 School of Civil Engineering and Architecture, Suqian University, Suqian 223800, China
4 Guizhou Water & Power Survey-Design Institute Co., Ltd., Guiyang 550001, China
* Correspondence: jtliu@hhu.edu.cn; Tel.: +86-025-8378-7803; Fax: +86-025-8378-6606

**Abstract:** Flow duration curves (FDCs) that represent streamflow regime function through an empirical relationship between the FDC parameters and basin descriptors are widely adopted for hydrologic applications. However, the applications of this method are highly dependent on the availability of observation data. Hence, it is still of great significance to explore the process controls of underpinning regional patterns on streamflow regimes. In this study, we developed a new regionalization method of FDCs to solve the problem of runoff prediction for ungauged mountainous basins. Five empirical equations (power, exponential, logarithmic, quadratic, and cubic) were used to fit the observed FDCs in the 64 mountainous basins in eastern China, and the power model outperforms other models. Stepwise regression was used to explore the differentiated control of 23 basin descriptors on the 13 percentile flows of FDCs, and seven descriptors remained as independent variables for further developing the regional FDCs. Application results with different combinations of these selected descriptors showed that five indices, i.e., average annual rainfall (P), average elevation (H), average gradient ($\beta$), average topographic index (TI), and maximum 7d of annual rainfall (Max7d), were the main control factors of FDCs in these areas. Through the regional method, we found that 95.31% of all the basins have *NSE* values greater than 0.60 and $\varepsilon$ (namely the relative mean square error) values less than 20%. In conclusion, our study can guide runoff predictions to help manage booming demands for water resources and hydropower developments in mountainous areas.

**Keywords:** flow duration curves; independent variables; regionalization; ungauged basin

## 1. Introduction

Mountainous areas are important headwater places with tremendous potential for hydropower generation, but sparse hydrological observations have greatly limited local water resources and hydropower developments [1–3]. Regionalization strategy, which transfers information (e.g., model parameters) from gauged (donor) basins to ungauged (receiver) basins, could provide a rational solution for PUB (Prediction in Ungauged Basins) problems [4–6]. Flow duration curves (FDCs) can provide a simple, comprehensive, and graphical characterization of basin runoff over the entire period for different scenarios (e.g., dry and wet), and are widely adopted by water resources management such as hydropower potential assessment, river sedimentation, and water quality management [7–9]. Moreover, FDCs regionalization requires a sound understanding and knowledge of hydrological processes and spatial-temporal heterogeneity of basin characteristics [10]. Hence, exploring the process controls underpinning regional patterns of streamflow regime behaviors is still worth revisiting [11–13]. In this study, we aim to develop a new regionalization method of FDCs to solve the problem of runoff prediction for ungauged mountainous basins.

According to Castellarin et al. [14], the available regionalization procedures to develop FDCs can be divided into three categories: graphical, statistical, and parametric approaches.

The graphical approach adopts standardized graphical representations of FDCs to develop regional dimensionless FDCs, while the latter two methods both depend on prior hypotheses of the distribution or shape of the regional FDCs [15,16]. In the statistical approaches, FDCs are generally considered as complementary curves of the daily streamflow cumulative distribution function (CDF) [7,10,17,18]. Furthermore, the parametric approach is based on either simple or complex empirical equations, whose parameters are calibrated according to the observed data using error minimization procedures [19,20]. However, FDCs cannot be interpreted as probability curves, because discharge is correlated between successive time intervals and discharge characteristics are dependent on the season [5]. Compared with the aforementioned two approaches, the parametric models, though possessing a more straightforward structure, do not necessarily lead to poorer performance [14,21].

Moreover, parameter regionalization of statistical and parametric models is usually based on regression models which relate the model parameters to basin descriptors [14,21]. This method assumes that basin descriptors shape the form of FDCs [22]. If provided with a presumed functional model and appropriate basin descriptors, the parametric regression can integrate a wide range of information and performs efficiently [10]. Thus, the primary challenge of this method is searching for the best empirical equation to fit the FDCs. In the parametric approach, five functions such as power, exponential, logarithmic, quadratic, and cubic functions are generally applied to describe FDCs [20,23,24]. However, the choice of functional model is not commonly agreed upon and is always region-specific [24,25]. In addition, most existing studies found that the exponential function or cubic function could accurately capture the characteristics of FDCs for basins with a substantial storage capacity [19,20,23,26], the FDCs of which have a slight overall slope and whose low-flow percentiles are large [27]. However, as pointed out by Ling et al. [25], the cubic function lacks an underlying hydrological concept because it may fluctuate in low-flow parts. Moreover, in headwater basins, the underlying surface conditions are complex and the streamflow process rises and falls sharply [28], where the FDCs are steeper in the high-flow percentiles and more minor in the low-flow percentiles. Thus, the form of the functions for headwater areas needs to be further examined through parametric approaches for practical applications.

The parametric approaches have shown that several basin descriptors dominate the shape of FDCs [29,30]. For instance, topographic features, including basin area (A), average elevation (H), and average gradient (β) have a strong influence on the shape of FDCs [19,26,31]. Yang et al. [32] emphasized the main controlling effect of rainfall features (e.g., average monthly rainfall and rainstorm features) on runoff response in eastern China. Smakhtin et al. [33] also demonstrated the role of average annual rainfall (P) on the shape of FDCs in South Africa. Cheng et al. [18] found that the aridity index (AI) cannot be ignored in controlling the FDCs in the arid areas of America. Furthermore, Ward and Robinson [34] highlighted the effects of soil and geology features on FDCs in the UK. Although the critical controls of FDCs in different regions could differ significantly, the consensus among hydrologists is that the shape of FDCs represents the collective impact of rainfall features, topographic features, and soil and geology features [11,29].

However, the empirical selection of geographical features would always lead to multicollinearity among independent variables, i.e., the performance of a small number of selected independent variables may be like that of many independent variables [22,35]. Jin et al. [36] also found that the inclusion of more indicators may not consistently produce better classification results. Moreover, the fewer independent variables selected, the easier it is to estimate FDCs through the regional model. Consequently, stepwise regression is always suggested to evaluate the significant impact of the descriptors on FDCs to avoid multicollinearity problems [37].

This paper aims to develop regional FDCs to solve the problem of runoff prediction in the ungauged mountainous basins of eastern China, and evaluate the performance of the model. We construct the regional FDCs based on geographical characteristics and improve the performance of the model through the optimal selection of the functional model and

independent variables. In addition, we explore the integrations of geographical controls on the regional streamflow regime function in the mountainous area.

## 2. Materials and Methods

### 2.1. Study Area and Data

Sixty-four small- to medium-sized basins (112°31′–120°55′ E, 25°30′–34°24′ N) located in eastern China were selected as the study area (Figure 1). Most of these basins are geographically contiguous, with close climates, landscape conditions, and hydrologic regimes (see Yang et al. [32] for specific information). The study area covers two climate zones, i.e., the warm temperate sub-humid monsoon climate to the north and the subtropical humid monsoon climate to the south. There are four distinct seasons, plenty of light, and the same period of rain and heat. The average annual temperature is 12–21 °C. The annual average precipitation is between 670 mm and 1973 mm, gradually decreasing from the southeast coast to the northwest inland. There are five prominent mountains, the Funiu Mountains, the Tongbai Mountains, the Dabie Mountains, the Tianmu Mountains, and the Wuyi Mountains, serving as the main flood source areas. The average elevation of the basin ranges from 55 m to 841 m, and the average gradient ranges from 4.27° to 27.13°.

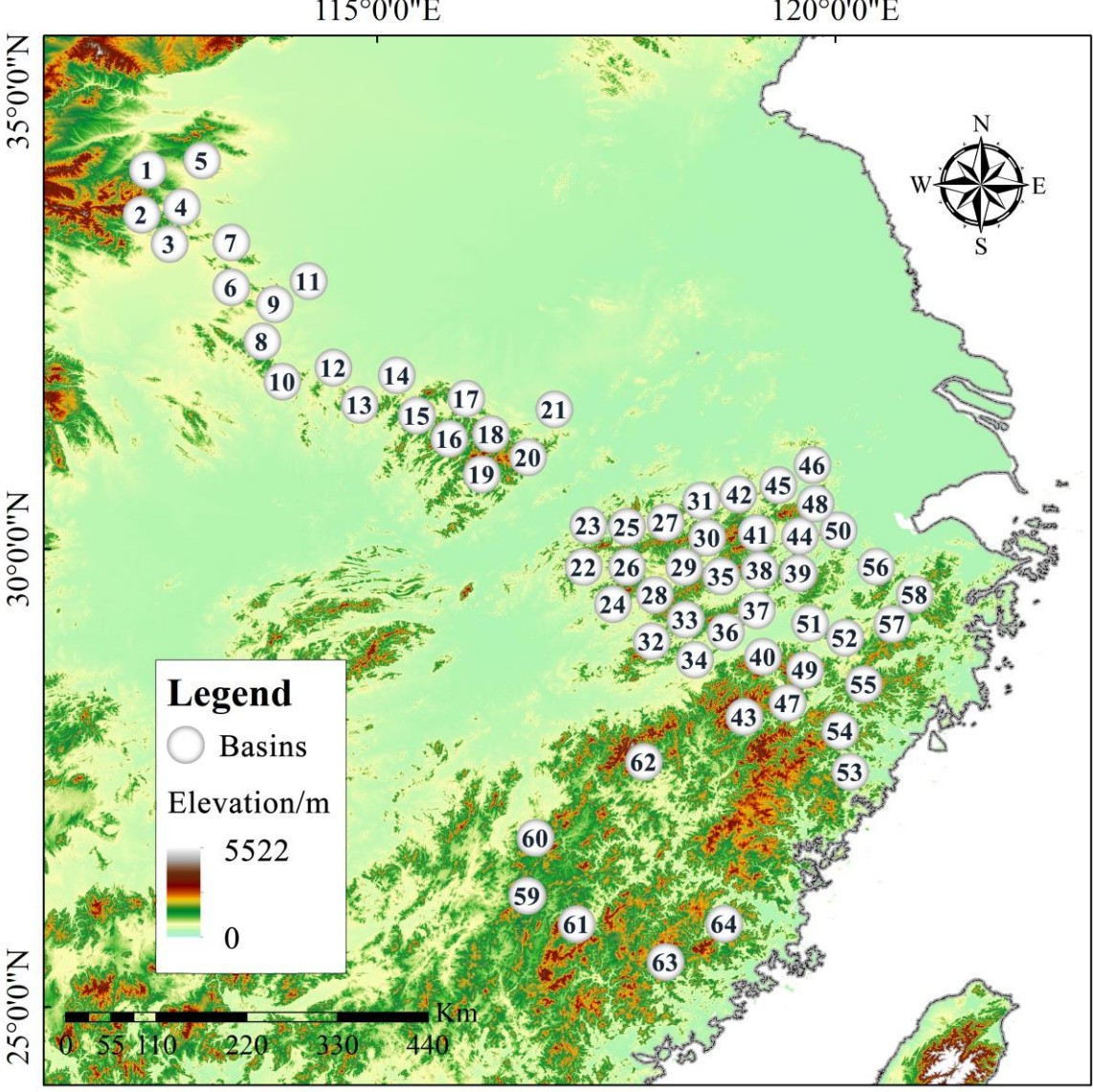

**Figure 1.** The spatial distribution of 64 mountainous basins used in the study.

In this paper, the daily rainfall and discharge data for each selected basin were obtained from the Hydrologic Yearbook of the People's Republic of China, and related hydrologic metrics were calculated by IHA (http://conserveonline.org/workspaces/iha, accessed on 1 August 2020). The 30 m Advanced Spaceborne Thermal Emission and Reflection Radiometer Global Digital Elevation Model (ASTER GDEM) and Landsat 4–5 TM land cover data were obtained from the Geographical Spatial Data Cloud of the Computer Network Information Center of the Chinese Academy of Sciences (http://www.gscloud.cn/, accessed on 5 October 2019). Soil types were obtained from the China Soil Scientific Database provided by the Institute of Soil Science of the Chinese Academy of Sciences (http://www.soil.csdb.cn/, accessed on 5 October 2019). The soil raster data were obtained from the Chinese Soil Dataset Based on the World Soil Database (HWSD) (http://www.crensed.ac.cn/portal, accessed on 5 October 2019), and the soil attributes were determined by the 'Soil Water Features' software (http://www.ars.usda.gov/ba/anri/hrsl/ksaxton, accessed on 5 October 2019). The geological data were obtained from the 1:1.5 million geology of the 'Geocloud' of the People's Republic of China (http://geocloud.cgs.gov.cn, accessed on 5 October 2019).

*2.2. Flow Duration Curve Method*

In this study, the total duration method is used to derive the area-normalized flow duration curves. Basin area, geographical location, availability, and quality of discharge data are considered when selecting basins. First, the flow is sorted for the entire period in descending order of magnitude. To compare the difference between FDCs in different basins, the actual flow value is standardized by the basin area. The percentile flow corresponding to discharge is estimated with the following formula:

$$Q_D = \frac{M}{N+1} \times 100 \tag{1}$$

where $Q_D$ is a percentile flow; $M$ is the assigned rank of the flow; $N$ is the total number of data points; and $D$ is the percentage of time that a given flow is met or exceeded. Here, 13 flow percentiles, i.e., $Q_1$, $Q_5$, $Q_{10}$, $Q_{20}$, $Q_{30}$, $Q_{40}$, $Q_{50}$, $Q_{60}$, $Q_{70}$, $Q_{80}$, $Q_{90}$, $Q_{95}$ and $Q_{99}$, are determined. For example, $Q_1$ represents the flow that equaled or exceeded 1% of the entire period of flood records (extremely high flow), and $Q_{99}$ represents the flow that equaled or exceeded 99% of the time (extremely low flow). Each flow percentile represents the different sections of FDCs.

Then, five mathematical models, i.e., the power model [Equation (2)], the exponential model [Equation (3)], the logarithmic model [Equation (4)], the quadratic model [Equation (5)], and the cubic model [Equation (6)], are tested to fit the FDCs.

$$Q_D = aD^{-b} \tag{2}$$

$$Q_D = ae^{(-bD)} \tag{3}$$

$$Q_D = a - b\ln D \tag{4}$$

$$Q_D = a - bD + cD^2 \tag{5}$$

$$Q_D = a - bD + cD^2 - dD^3 \tag{6}$$

where $a$, $b$, $c$, and $d$ are the parameters resulting from the curve fit, and they are all positive. A total of 13 flow percentiles, including $Q_1$, $Q_5$, $Q_{10}$, $Q_{20}$, $Q_{30}$, $Q_{40}$, $Q_{50}$, $Q_{60}$, $Q_{70}$, $Q_{80}$, $Q_{90}$, $Q_{95}$ and $Q_{99}$, yield 13 pairs of ($Q_D$, $D$) values. Then, the functional models are fitted to each set of 13 pairs of ($Q_D$, $D$) values for each basin using the least-squares method.

*2.3. Independent Variables Selection*

Following Yang et al. [32], 23 basin descriptors are considered, including 6 rainfall features, 12 topographic features, and 5 soil and geological features. Detailed descriptions can be

found in Table 1. The concrete values of 23 descriptors in 64 study basins are shown in Tables A1–A3. Because of the multicollinearity among the 23 basin descriptors selected, stepwise regression [38] is performed to analyze the significant impacts of the descriptors on the 13 percentile flows in the above-selected basins. It determines the predictors (basin descriptors) individually and identifies a set of predictors with the lowest Akaike Information Criterion (AIC). The selected set of basin descriptors for each flow percentile significantly affects that flow percentile, while the removed ones are not significant enough. Finally, the control of basin descriptors on the 13 percentile flows is revealed and the influential ones are selected as independent variables for the following work.

**Table 1.** Description of the 23 basin descriptors.

|  | Abbreviation | Description | Unit |
|---|---|---|---|
| Rainfall | P | Average annual rainfall | mm |
|  | Max 1d | Maximum 1 d of annual rainfall | mm |
|  | Max 3d | Maximum 3 d of annual rainfall | mm |
|  | Max 7d | Maximum 7 d of annual rainfall | mm |
|  | RR | Rate of increase in daily rainfall [a] | mm/d |
|  | FR | Rate of decline in daily rainfall [b] | mm/d |
| Topography | A | Basin area | $km^2$ |
|  | H | Average elevation | m |
|  | $\beta$ | Average gradient | $\circ$ |
|  | CC | Average plan curvature [c] | - |
|  | CP | Average profile curvature [d] | - |
|  | TI | Average topographic index | - |
|  | HI | Integral of area–altitude curve | - |
|  | AS | Gradient of integral of area–altitude curve | - |
|  | RC | Circularity ratio | - |
|  | RF | Form factor | - |
|  | D | Fractal dimension of river network | - |
|  | NDVI | Normalized difference vegetation index | - |
| Soil and geology | AW | Available moisture content [e] | mm/m |
|  | SHC | Saturated hydraulic conductivity | cm/s |
|  | MBD | Matric bulk density | $g/cm^3$ |
|  | IR | Proportion of impermeable rock formation (igneous rock) area | % |
|  | FLD | Line density of cracks or faults | $km/km^2$ |

Note(s): [a] The average of all positive differences between consecutive days. [b] The average of all negative differences between consecutive days. [c] Positive values indicate divergent gradients, and negative values indicate convergent gradients. [d] The second derivative of altitude or surface elevation and a positive value indicates a convex gradient, and a negative value indicates a concave gradient. [e] According to the soil raster data source, the uniform average soil thickness is 1 m.

### 2.4. Regionalization Approach

Regional models are constructed utilizing multiple regression among the parameters (*a*, *b*, *c* and *d*) defined in the fit phase [Equations (2)–(6)] with the selected basin descriptors. Since the values of parameters *a*, *b*, *c*, and *d* are all positive, nonlinear regression is used. The regression equation applied here is as follows:

$$V = \beta_0 \prod_{j=1}^{n} \alpha_j^{\beta_j} \tag{7}$$

where *V* is the vector that represents parameters *a*, *b*, *c* and *d*; $\beta_0$ is a regression constant; $\beta_j$ is the regression coefficients; $\beta_0$ and $\beta_j$ are determined by the least-squares method; and $\alpha_j$ is the independent variables, i.e., the basin descriptors selected in the previous section. *n* is the number of independent variables.

*2.5. Leave-One Cross-Validation Method*

The leave-one cross-validation method is adopted to validate the regionalization models. The principle of cross-validation is to divide the raw data into different small subsets; it then analyses one small subset first and verifies the other small subsets later. The leave-one cross-validation first selects one basin as the validation basin and the remaining basins as the reference basins. Then, it establishes a regionalization model of FDCs through the reference basins' data, to deduce the FDCs of the validation basin and compare it with the observed FDCs. This process is repeated until all basins end up as the validation basins. Thus, the simulated percentile flow $Q_{S,D_i}$ and the observed percentile flow $Q_{O,D_i}$ of the validation basin under the specified relative duration $D$ can be obtained. For each basin, we conduct its model development on the data of the other 63 basins and validate it on its own data. The leave-one cross-validation is assessed by the Nash–Sutcliffe efficiency (*NSE*) coefficient [39] and the relative mean square error ($\varepsilon$), which are described as follows:

$$NSE = 1 - \frac{\sum_{i=1}^{n}\left(Q_{S,D_i} - Q_{O,D_i}\right)^2}{\sum_{i=1}^{n}\left(Q_{O,D_i} - \overline{Q_{O,D_i}}\right)^2} \tag{8}$$

$$\varepsilon(\%) = \frac{1}{n}\sqrt{\sum_{i=1}^{n}\left(\frac{Q_{O,D_i} - Q_{S,D_i}}{Q_{O,D_i}}\right)^2} \times 100 \tag{9}$$

where $Q_{O,D_i}$ is the observed percentile flow; $\overline{Q_{O,D_i}}$ is the average observed percentile flow; $Q_{S,D_i}$ is the simulated percentile flow; and $n$ is the length of the data. The *NSE* ranges from $-\infty$ to 1.0, with 1.0 representing the best fit [39]. The optimal value of $\varepsilon$ is zero.

## 3. Results

*3.1. The Selection of Functional Models*

Following the key steps described in Section 2.2, the optimal model was selected from the five functional models after constructing the FDCs for the 64 mountainous basins in eastern China using the total duration method. Figure 2 shows the box charts of the *NSE* values and the $\varepsilon$ values between the fitted and observed FDCs with five models for the 64 basins. The power function has the highest *NSE* value and the lowest $\varepsilon$ value among all the functions, followed by the exponential function. In the results of these two functions, the average *NSE* values of all 64 basins are 0.99 and 0.97, and their average $\varepsilon$ values are only 0.76% and 1.22%, respectively. The quadratic function has the worst-fitting results with an average *NSE* value of 0.65 and $\varepsilon$ value of 4.54%. There are even two basins with *NSE* values of less than 0.50. The average *NSE* values of the logarithmic and cubic functions are 0.84 and 0.79, which are much lower than those of the power and exponential function, so the power function is defined as the best model for the study area. The best-fit parameters $a$, and $b$ values for the power function and their *NSE* values, and $\varepsilon$ values for each basin are listed in Table 2. Parameters $a$ and $b$ represent geographical and climatic information that affect flows and will be transferred to regionalization. When using the power function, all 64 basins are well-fitted with *NSE* values greater than 0.97 and $\varepsilon$ values lower than 1.66%. There are three basins with *NSE* values of 0.97, eleven basins of 0.98, thirty-one basins of 0.99, and others of 1.00.

Figure 3 shows the five fitting equations of eight basins to the observed FDCs. It can be deduced that the hydrograph of the quadratic function starts to rise in the low-flow percentiles, and the hydrograph of the cubic function fluctuates in the streamflow exceeded $Q_{40}$. The logarithmic function always underestimates $Q_1$ and the low-flow percentiles ($Q_{70}$–$Q_{99}$) and overestimates other flow percentiles ($Q_5$–$Q_{50}$). It also shows that both the power and the exponential function can satisfy the shape of FDCs, especially in extremely-high-flow percentiles. However, the exponential function always underestimates the flow percentiles of $Q_{10}$–$Q_{50}$ by a wide margin. Therefore, the power function is the best model to capture the shapes of FDCs in mountainous areas.

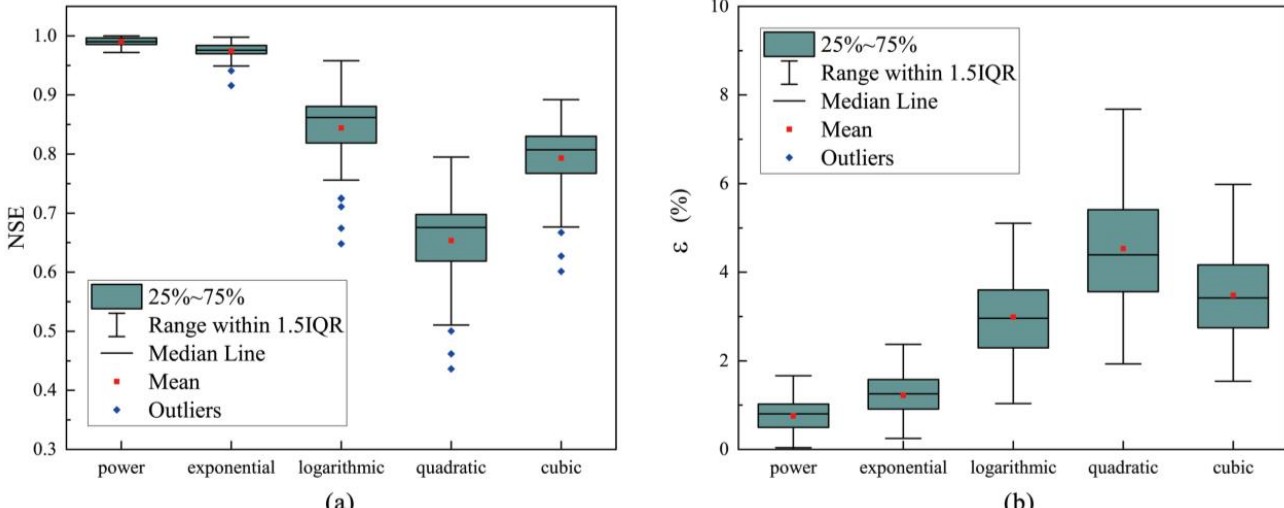

**Figure 2.** The box charts of (**a**) the Nash–Sutcliffe efficiency (*NSE*) and (**b**) the relative mean square error (*ε*) values between the fitted FDCs and the observed FDCs for the 64 basins with the five models, including power, exponential, logarithmic, quadratic and cubic models.

**Table 2.** Parameters *a* and *b* of the power model and the *NSE* and *ε* values in the FDC fit for each basin.

| ID | Station | *a* | *b* | *NSE* | *ε* (%) | ID | Station | *a* | *b* | *NSE* | *ε* (%) |
|----|---------|-----|-----|-------|---------|----|---------|-----|-----|-------|---------|
| 1 | Ziluoshan | 0.0016 | 0.9202 | 1.00 | 0.10 | 33 | Zhongzhou | 0.0126 | 0.7578 | 0.99 | 1.18 |
| 2 | Zhongtang | 0.0023 | 0.9659 | 1.00 | 0.23 | 34 | Yancun | 0.0143 | 0.7543 | 0.99 | 1.34 |
| 3 | Jizhong | 0.0036 | 0.9263 | 1.00 | 0.33 | 35 | Baikuoban | 0.0104 | 0.7582 | 0.99 | 0.88 |
| 4 | Xiagushan | 0.0013 | 1.0352 | 1.00 | 0.07 | 36 | Shangzouban | 0.0123 | 0.6882 | 0.98 | 1.11 |
| 5 | Gaocheng | 0.0066 | 0.8471 | 1.00 | 0.34 | 37 | Yuankou | 0.0107 | 0.7387 | 0.99 | 1.06 |
| 6 | Lixin | 0.0010 | 1.2433 | 1.00 | 0.05 | 38 | Qingshandian | 0.0127 | 0.6941 | 0.98 | 1.11 |
| 7 | Guanzhai | 0.0011 | 1.0302 | 1.00 | 0.06 | 39 | Fenshui | 0.0110 | 0.7129 | 0.98 | 0.99 |
| 8 | Dapoling | 0.0025 | 0.9074 | 1.00 | 0.19 | 40 | Shanjiao | 0.0172 | 0.6387 | 0.97 | 1.50 |
| 9 | Luzhuang | 0.0011 | 1.0941 | 1.00 | 0.09 | 41 | Laoshikan | 0.0109 | 0.6184 | 0.97 | 0.87 |
| 10 | Tanjiahe | 0.0036 | 0.9749 | 1.00 | 0.35 | 42 | Qianyu | 0.0076 | 0.7491 | 0.99 | 0.75 |
| 11 | Zhumadian | 0.0003 | 1.3868 | 1.00 | 0.04 | 43 | Shangbao | 0.0117 | 0.7194 | 0.99 | 1.02 |
| 12 | Nanlidian | 0.0030 | 0.9483 | 1.00 | 0.26 | 44 | Qiaodongcun | 0.0087 | 0.7359 | 0.99 | 0.70 |
| 13 | Peihe | 0.0050 | 0.9219 | 1.00 | 0.57 | 45 | Hengtangcun | 0.0096 | 0.6685 | 0.99 | 0.57 |
| 14 | Baiqueyuan | 0.0047 | 0.9019 | 1.00 | 0.40 | 46 | Yubujie | 0.0079 | 0.7217 | 0.98 | 0.81 |
| 15 | Huangnizhuang | 0.0052 | 0.8705 | 1.00 | 0.36 | 47 | Shangxiantan | 0.0100 | 0.7494 | 0.99 | 1.00 |
| 16 | Qilin | 0.0075 | 0.8524 | 1.00 | 0.54 | 48 | Jiangwan | 0.0091 | 0.7549 | 0.98 | 1.11 |
| 17 | Zhangchong | 0.0076 | 0.7999 | 1.00 | 0.46 | 49 | Liantangkou | 0.0085 | 0.7229 | 0.98 | 0.82 |
| 18 | Bailianya | 0.0076 | 0.8118 | 1.00 | 0.50 | 50 | Daixi | 0.0071 | 0.7569 | 0.98 | 0.79 |
| 19 | Huanghewei | 0.0129 | 0.7189 | 1.00 | 0.60 | 51 | Yiwufotang | 0.0078 | 0.7232 | 0.99 | 0.61 |
| 20 | Xiaotian | 0.0096 | 0.7869 | 0.99 | 0.79 | 52 | Dongyangyanxia | 0.0068 | 0.7666 | 0.99 | 0.62 |
| 21 | Taoxi | 0.0055 | 0.7771 | 0.99 | 0.59 | 53 | Daitou | 0.0140 | 0.7616 | 0.99 | 1.03 |
| 22 | Liukou | 0.0163 | 0.7470 | 0.99 | 1.33 | 54 | Qiulu | 0.0125 | 0.7516 | 0.99 | 0.85 |
| 23 | Dahekou | 0.0114 | 0.7862 | 0.99 | 1.01 | 55 | Caodian | 0.0110 | 0.7451 | 0.99 | 1.11 |
| 24 | Yuetan | 0.0145 | 0.7214 | 0.99 | 1.21 | 56 | Shuangjiangxi | 0.0084 | 0.7574 | 0.98 | 0.99 |
| 25 | Shancha | 0.0167 | 0.7178 | 0.98 | 1.66 | 57 | Xixi | 0.0095 | 0.7320 | 0.99 | 0.79 |
| 26 | Wananba | 0.0114 | 0.7245 | 0.99 | 0.91 | 58 | Huangze | 0.0064 | 0.7766 | 0.99 | 0.50 |
| 27 | Sankouzhen | 0.0164 | 0.6811 | 0.98 | 1.47 | 59 | Yutan | 0.0132 | 0.6616 | 0.99 | 0.93 |
| 28 | Tunxi | 0.0139 | 0.7177 | 0.99 | 1.26 | 60 | Jianning | 0.0147 | 0.6319 | 0.99 | 0.81 |
| 29 | Yuxi | 0.0089 | 0.7803 | 0.99 | 0.72 | 61 | Hongtian | 0.0151 | 0.5558 | 0.97 | 0.89 |
| 30 | Linxi | 0.0077 | 0.7625 | 0.99 | 0.64 | 62 | Chongan | 0.0148 | 0.6985 | 0.99 | 0.85 |
| 31 | Hulesi | 0.0097 | 0.7491 | 0.99 | 0.85 | 63 | Fengyang | 0.0169 | 0.6194 | 0.98 | 1.08 |
| 32 | Misai | 0.0135 | 0.7834 | 0.99 | 1.21 | 64 | Taipingkou | 0.0108 | 0.6970 | 0.99 | 0.73 |

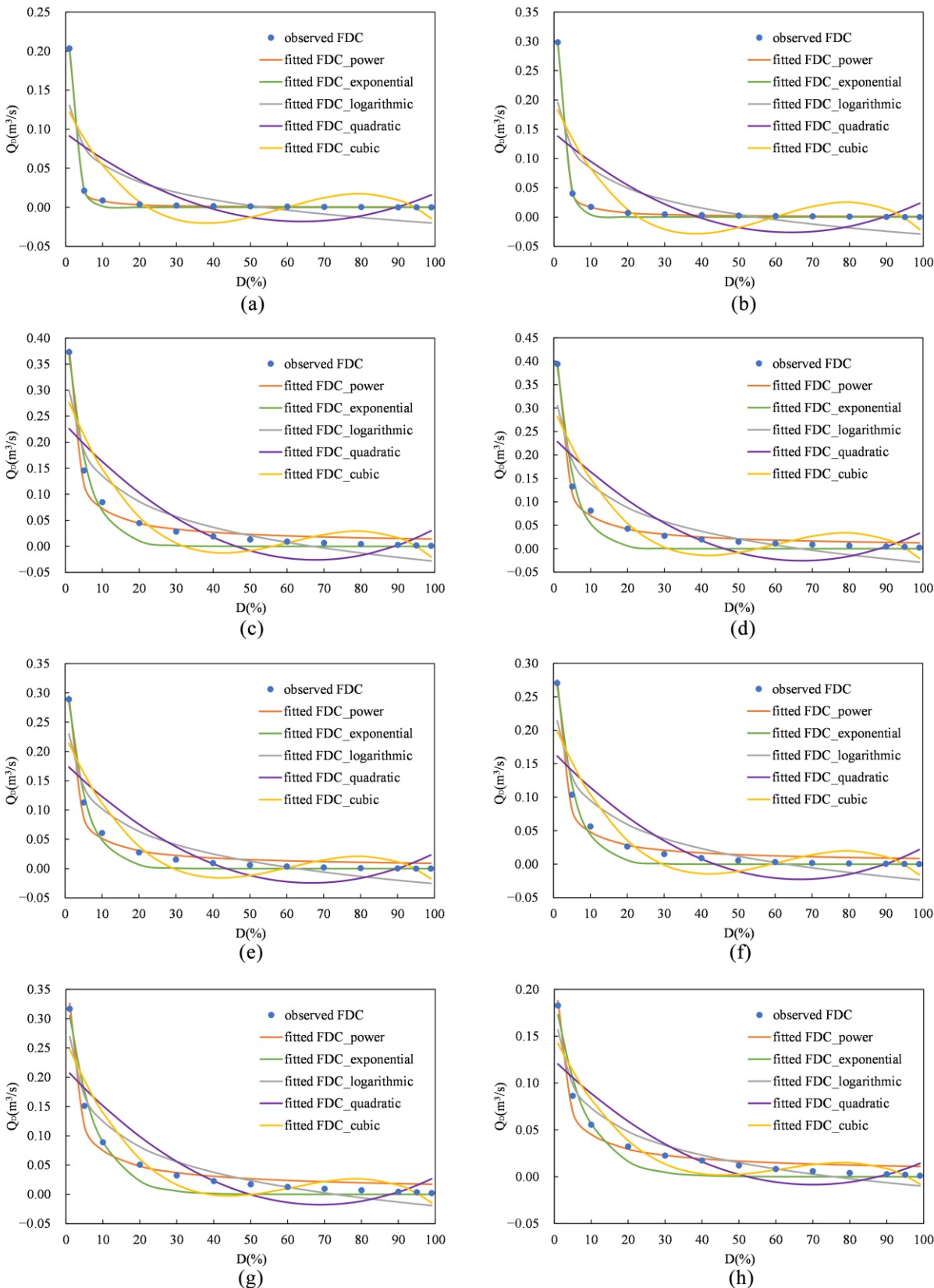

**Figure 3.** Five fitting equations of FDCs of eight basins from different *NSE* values: (**a**) Zhumadian (ID. 11 in Figure 1 & Table 2), (**b**) Lixin (ID. 6), (**c**) Tunxi (ID. 28), (**d**) Qiulu (ID. 54), (**e**) Jiangwan (ID. 48), (**f**) Shuangjiangxi (ID. 56), (**g**) Shanjian (ID. 40), and (**h**) Laoshikan (ID. 41).

### 3.2. Controlling Descriptors Adopted as Independent Variables

We first analyzed the control of basin descriptors on the 13 percentile flows to choose the best independent variables by performing stepwise regression. Figure 4 shows the results of stepwise regression of 23 basin descriptors on 13 percentile flows respectively. Here, positive standardized regression coefficients represent positive correlation and negative coefficients mean negative correlation, and the larger the absolute value, the more significant the effect. A total of 11 basin descriptors remained through the regression procedure, including average annual rainfall (P), average elevation (H), average gradient ($\beta$), average topographic index (TI), normalized difference vegetation index (NDVI), maximum 7d of annual rainfall (Max7d), maximum 3d of annual rainfall (Max3d), rate of decline in daily rainfall (FR), plan curvature (CC), saturated hydraulic conductivity (SHC) and basin area (A). The main controls for FDCs were the rainfall features that were more influential for the high-flow percentiles, followed by the topographic features. For the low-flow percentiles, topographic features were more effective. Almost all the soil and geological features except SHC were removed.

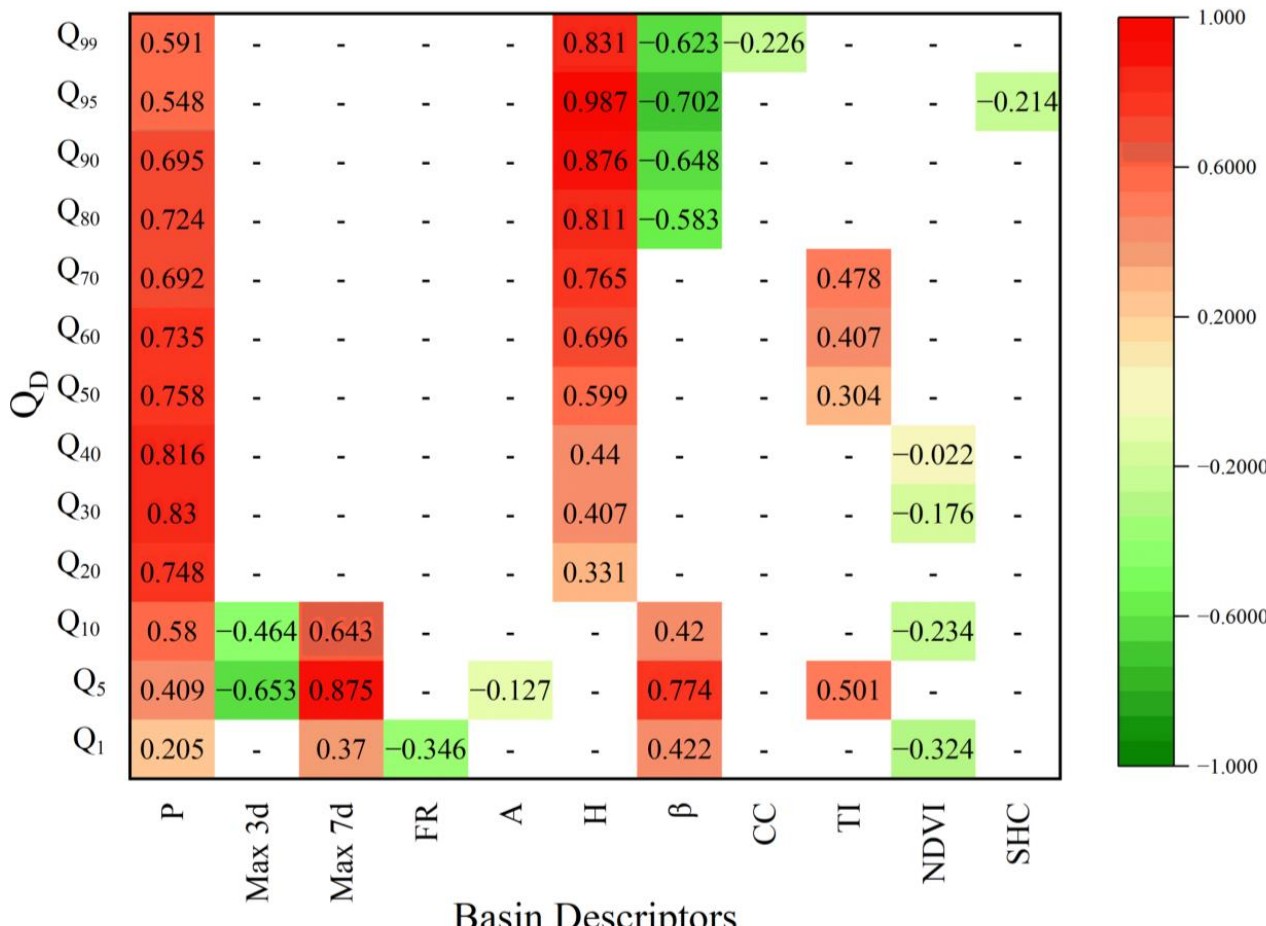

**Figure 4.** Results of the stepwise regression. The stepwise regression was performed on 23 basin descriptors for each of the 13 percentile flows and only 11 descriptors remained. The numbers represent standardized regression coefficients. No value (-) means that the basin descriptor is not significant enough for this percentile flow and is removed.

P, H, $\beta$ were the main control features for the FDCs. Of these, P was selected for all 13 flow percentiles, H was removed from only 3 extremely-high-flow percentiles ($D \leq 10\%$), and $\beta$ was selected in the regression analysis for 7 percentile flows. In addition, TI, NDVI, Max7d and Max3d also have certain control effects on the FDCs, being selected by 4, 4, 3 and 2 flow percentiles, respectively. Finally, it was found that streamflow strongly correlated

with the basin area [23]. However, in this study, we normalized the streamflow with basin area, so only 1 flow percentile was selected in the stepwise regression procedure when the basin area was performed. Very high flows are more sensitive to basin descriptors and their regressions incorporate more descriptors than other percentile flows. Additionally, the contribution of Max7d was more vital than that of P. This might be because high flows in small basins in the mountainous region always occur after heavy rainfall. Moreover, P, Max 7d, H, and TI have a positive effect on FDCs, and Max 3d, FR, A, CC, NDVI, and SHC have a negative effect on FDCs. Additionally, β has a positive effect on high flows and a negative effect on low flows. The former may be because steeper basins are more prone to flooding, and the latter may be because steeper basins have poorer basin storage capacity.

Above all, the basin descriptors P, H, β, and TI are the four most critical influencing attributes, so they were directly selected as independent variables in the regression. The basin descriptors FR, CC, SHC, and A were removed from the independent variables because they were all selected by only one percentile flow in the regression analysis. The other three basin descriptors, NDVI, Max7d, and Max3d, significantly influence the FDCs. Therefore, seven basin descriptors, P, H, β, TI, NDVI, Max7d, and Max3d, may be the best independent variables of regional FDCs, but their practical application still needs further evaluation.

### 3.3. Assessments of Regionalization Models

Based on the above analysis, these seven basin descriptors were divided into 4–7 independent variables for multiple regression analysis of model parameters. Three specific combinations of these seven descriptors are shown in Table 3. We first used a power function to fit the observed FDCs and obtained the estimated parameters *a* and *b* for each basin (Table 2). Then, the selected descriptors in different combinations were used as independent variables to establish regression relationships with parameters *a* and *b* separately. The best regionalization models for the estimated parameters *a* and *b* are listed in Table 3. Next, leave-one cross-validation was used to transfer the regionalization models to each basin to establish the regional FDCs. Different simulation results (*NSE* and *ε* values) were obtained according to seven different combinations of independent variables as shown in Figure 5. In combination I, only four basin descriptors, P, H, β, and TI, were used as independent variables. The mean and median *NSE* values of the 64 basins were 0.90 and 0.96, respectively. There were only three basins with *NSE* less than 0.50, and the value of *NSE* less than 0.50 was considered weak, fitting between the simulated and the observed curves [40–42]. On this basis, NDVI, Max7d, and Max3d were stepwise added as independent variables: combinations II, III and IV. Although combination II reduced the number of unqualified basins to two, the mean and median values of *NSE* obtained by the 64 basins were lower than those of combination I, and the overall simulation results were not improved. Combinations III and IV increased the mean *NSE* of 64 basins to 0.93 and 0.92, respectively. Combination III also reduced the number of unqualified basins to 2. Therefore, Combination III was selected as the best regressor of the 5-factor combination. Then, based on combination III, NDVI and Max3d were added as the independent variables of the 6-factor combination, namely, combinations V and VI. Finally, all seven basin descriptors were used as the independent variables, namely, combination VII, to establish the regionalization models, and cross-validation was performed. The mean values of *NSE* of combinations IV, V and VII in 64 basins were 0.92, 0.93, and 0.93, respectively, which were not significantly improved compared with combination III. There was no significant difference in the *ε* values of all 7 combinations – that is, there were always two basins that were greater than 20%, and their mean and median values were all 16% and 15% respectively. Although using the independent variables of combinations III and VI can produce similar overall simulation results in 64 basins, considering the simplicity in practice, five basin descriptors P, H, β, TI and Max7d in the combination III are selected as independent variables. This result indicates that only five basin descriptors, P, H, β, TI and Max7d, can well explain the FDCs in the eastern China mountainous areas.

**Table 3.** Different combinations of independent variables and their regionalization models.

| Combination | Independent Variables | Regionalization Models |
|---|---|---|
| I | P, H, β, TI | $a = 5.4460E - 17 \times P^{1.7804} \times H^{0.0976} \times \beta^{2.1535} \times TI^{6.8594}$ <br> $b = 3769.8 \times P^{-0.3189} \times H^{-0.0352} \times \beta^{-0.5584} \times TI^{-2.3221}$ |
| II | P, H, β, TI, NDVI | $a = 5.1758E - 17 \times P^{1.8341} \times H^{0.0757} \times \beta^{2.2039} \times TI^{6.5812} \times NDVI^{-1.8014}$ <br> $b = 3773.6 \times P^{-0.3199} \times H^{-0.0348} \times \beta^{-0.5594} \times TI^{-2.3166} \times NDVI^{0.0357}$ |
| III | P, H, β, TI, Max7d | $a = 1.6503E - 16 \times P^{1.9159} \times H^{0.1366} \times \beta^{2.0357} \times TI^{6.4075} \times Max7d^{-0.3412}$ <br> $b = 1392.0 \times P^{-0.4406} \times H^{-0.0703} \times \beta^{-0.4526} \times TI^{-1.9160} \times Max7d^{0.3066}$ |
| IV | P, H, β, TI, Max3d | $a = 8.6183E - 16 \times P^{1.9792} \times H^{0.1639} \times \beta^{1.8663} \times TI^{5.8928} \times Max3d^{-0.5038}$ <br> $b = 745.8 \times P^{-0.4355} \times H^{-0.0741} \times \beta^{-0.3899} \times TI^{-1.7549} \times Max3d^{0.2956}$ |
| V | P, H, β, TI, NDV I, Max7d | $a = 1.5009E - 16 \times P^{1.9486} \times H^{0.1184} \times \beta^{2.0790} \times TI^{6.2254} \times NDVI^{-1.3303} \times Max7d^{-0.3236}$ <br> $b = 1351.1 \times P^{-0.4303} \times H^{-0.0760} \times \beta^{-0.4390} \times TI^{-1.9734} \times NDVI^{-0.4189} \times Max7d^{0.3122}$ |
| VI | P, H, β, TI, Max7d, Max3d | $a = 1.9109E - 14 \times P^{1.8368} \times H^{0.1566} \times \beta^{1.5603} \times TI^{5.2272} \times Max7d^{2.2409} \times Max3d^{-2.3975}$ <br> $b = 622.4 \times P^{-0.4272} \times H^{-0.0737} \times \beta^{-0.3721} \times TI^{-1.7161} \times Max7d^{-0.1307} \times Max3d^{0.4061}$ |
| VII | P, H, β, TI, NDVI, Max7d, Max3d | $a = 2.0231E - 14 \times P^{1.8283} \times H^{0.1612} \times \beta^{1.5464} \times TI^{5.2629} \times NDVI^{0.3224} \times Max7d^{2.2551} \times Max3d^{-2.4147}$ <br> $b = 547.6 \times P^{-0.4082} \times H^{-0.0839} \times \beta^{-0.3409} \times TI^{-1.7961} \times NDVI^{-0.7232} \times Max7d^{-0.1627} \times Max3d^{0.4447}$ |

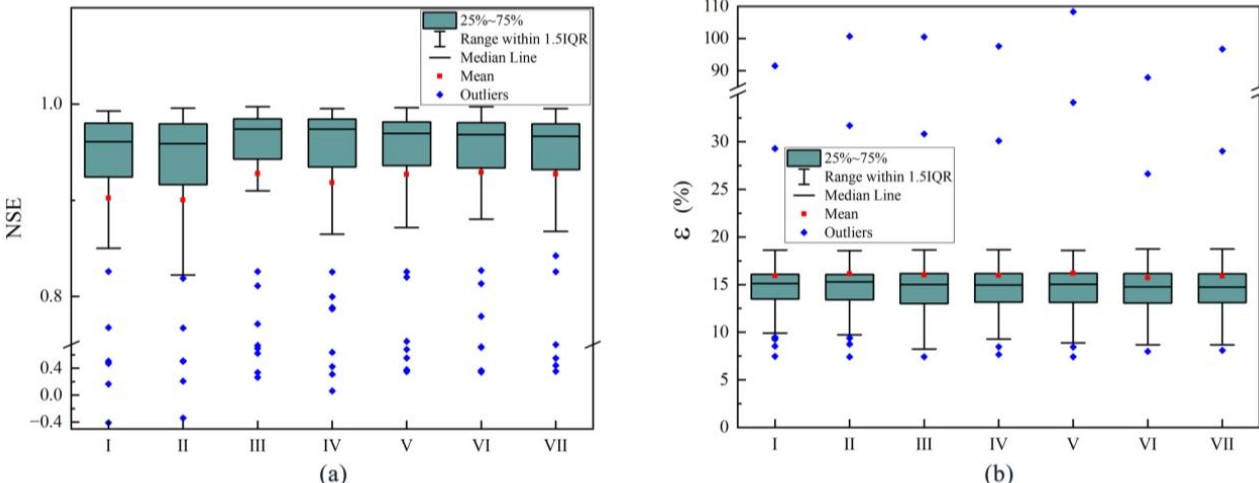

**Figure 5.** The box charts of (**a**) the *NSE* values and (**b**) the ε values of FDCs obtained by simulation for different combinations.

Table A4 shows the values of *NSE* and ε the FDCs obtained from the leave-one-out cross-validation of the 64 basins under the combination III. Only 2 of the 64 basins had an *NSE* less than 0.50, namely Ziluoshan (ID. 1) and Gaocheng (ID. 5), so they were unsatisfactory. The remaining basins all had *NSE*s greater than 0.60. There were six basins with *NSE* values between 0.60–0.90, including two basins (ID. 41 & 61) between 0.6–0.7, two basins (ID. 2 & 4) between 0.7–0.8, and another two basins (ID. 45 & 6) between 0.80–0.90. The values of *NSE* in the remaining 56 basins are all above 0.90, accounting for 87.5% of the total. The average *NSE* for all basins was 0.93. For ε, only two basins had an ε of more than 20%, namely Gaocheng (ID. 5) and Taoxi (ID. 21). All 64 basins had an average ε of 16%, indicating that the model was generally effective. For example, Figure 6 shows eight simulated basins to the observed FDCs, with every two basins within different *NSE* gradient ranges (0.90–1.00, 0.80–0.90, 0.70–0.80, 0.60–0.70).

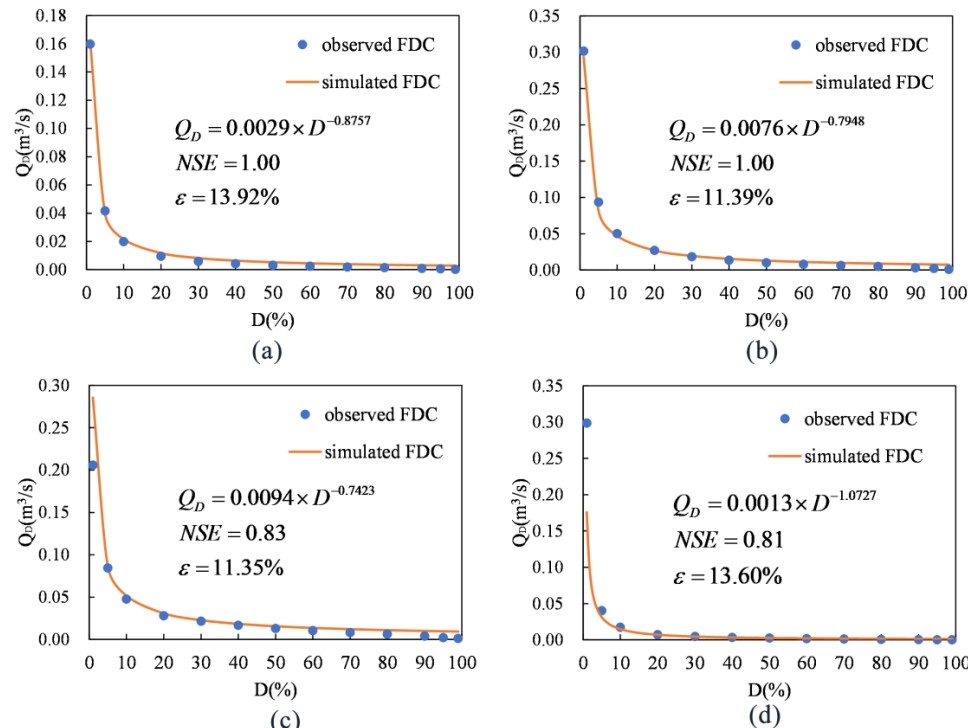

**Figure 6.** *Cont.*

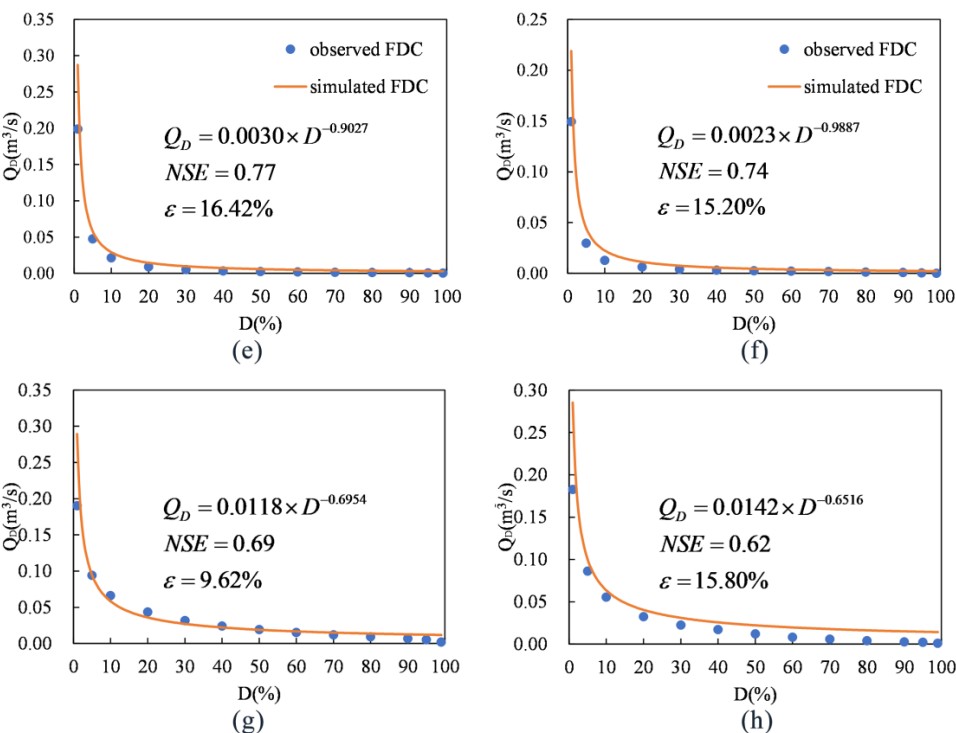

**Figure 6.** The simulated and observed FDCs of eight basins from different *NSE* gradient ranges:
(**a**) Dapoling (ID. 8), (**b**) Zhangchong (ID. 17), (**c**) Hengtangcun (ID. 45), (**d**) Lixin (ID. 6), (**e**) Zhongtang
(ID. 2), (**f**) Xiagushan (ID. 4), (**g**) Laoshikan (ID. 41) and (**h**) Hongtian (ID. 61).

## 4. Discussions

### 4.1. Applicability of the Model

It can be observed that the power model was the most appropriate for predicting flow
behaviors in the studied basins. However, previous studies rarely chose power functions to
build the regional models, in which cubic [20,23,24], exponential [19,26,43], and logarithmic
models [15] were usually used. According to Silva et al. [23], this difference is because each
basin's physical and climatic characteristics are unique. The optimal fitting model differs
for regions [24] and percentile flows [25]. In our study area, characterized by steep rise
and fall of the flood hydrographs in the headwaters, the FDCs in the high-flow percentiles
are incredibly steep. At first glance in Figure 3, the quadratic function starts to rise in the
low-flow percentiles, and the hydrograph of the cubic function fluctuates in the streamflow
exceeded $Q_{40}$. These results are consistent with the study by Ling et al. [25] that polynomial
functions fail to capture and align the observed FDCs. The logarithmic function seems
more suitable for describing the FDCs of basins with higher storage capacity [26], because
it always underestimates $Q_1$ and the low-flow percentiles ($Q_{70}$–$Q_{99}$) and overestimates
other flow percentiles ($Q_5$–$Q_{50}$) in the study area. Both the power and the exponential
function satisfy the shape of FDCs in which the gradient in the high-flow percentiles is
high, while it is relatively gentle in the low-flow percentiles. Nevertheless, the latter always
underestimates the flow percentiles of $Q_{10}$–$Q_{20}$ by a wide margin. Therefore, in this study,
the shape of the power function can best match the FDCs in humid mountainous areas.

It was also found that the critical controls for FDCs were the rainfall features and the to-
pographic features. These results are also in agreement with those found by Yang et al. [32],
in which rainfall features had the vital control over the hydrological responses of half of
the basin groups, followed by the topographic features. We also found that only five basin
descriptors of P, H, β, TI, and Max7d can well explain the FDCs in the mountainous areas.
However, previous studies have shown that basin area (A) was also commonly used as
an independent variable in regionalization models [23,24,44,45]. The widely-used basin
area (A) was not explicitly adopted as an independent variable because A was used to

normalize the streamflow in this study. In addition, TI and Max7d were also selected as independent variables. Jin et al. [36] found that average elevation (H), average gradient (β), and average topographic index (TI) have the best indication of hydrologic response to high flow. Therefore, the choice of TI in this region is understandable. Because the high flows of small basins in mountainous areas mainly occurred after heavy rainfalls, Max7d was selected as an independent variable of the regression model, and it was a strong indicator of the high-flow percentiles in the study area.

From the assessment results, it can be concluded that 95.31% of the basins have values of *NSE* greater than 0.60 and ε less than 20%. According to Pessoa et al. [24] and Silva et al. [23], regionalization models presented a satisfactory performance in the estimation of FDCs when the average value of *NSE* was higher than 0.60 and ε was less than 20%. Hence, our results indicate that there is a satisfactory fit between the observed FDCs and the FDCs simulated by the model in more than 95% of the cases. In conclusion, the regional FDCs we developed are suitable for mountainous areas.

### 4.2. Future Research Questions

Regionalization requires an understanding of hydrological processes and the spatiotemporal heterogeneity of climate and landscape features [10,11,18]. The simple nearest-neighbor method to define the similarity between basins is insufficient [46,47]. In this study, we considered the entire study area a single homogeneous region, given the strong similarities among the empirical FDCs. However, there are still two basins (i.e., Ziluoshan [ID. 1] and Gaocheng [ID. 5]) that could not be satisfactorily simulated no matter which combination of independent variables was used. They cannot be considered homogeneous with most of the other 62 basins. Yang et al. [32] proposed a stepwise clustering scheme to classify these 64 mountainous basins into 11 basin groups and three hierarchies; Ziluoshan (ID. 1) and Gaocheng (ID. 5) are classified into the same group with the highest similarity. Analyzing their basin descriptors (Tables A1–A3), we found that all the rainfall features were well below the average of the 64 basins. For example, P, Max 1d and Max 3d of Ziluoshan (ID. 1) were 48.49%, 33.18%, and 26.43% lower than the average values of the donor basins, and those of Gaocheng (ID. 5) were 54.63%, 36.59%, 39.37% lower than the average values of the donor basins, respectively. We also found that a significant deviation of P between the receiver basin and donor basins would generally suggest this receiver basin is heterogeneous from the donor basins. However, although the P of Xiagushan (ID. 4) and Guanzhai (ID. 7) were 48.22% and 44.80% lower than the average of donor basins, they still have a satisfactory simulation. This may be because the deviations of their Max 1d and Max 3d between the receiver basin and donor basins were only 11.96% and 18.10%, respectively. Deviations in P and Max 1d values greater than 40% and 20% between the receiver basin and donor basins may be an indication of heterogeneity. According to Boscarello et al. [40], the performance of the regional FDCs can be greatly improved by classifying homogeneous regions. Hence, the definition of homogeneous regions is essential for regionalization. Our results may guide the subsequent identification of homogenous regions. In addition, more donor basins need to be added to ensure enough similar alternative reference basins for these two basins. Subsequent studies can be combined with hydro-climatic classification or basin classification [48–50], including more basins to improve the performance of the regionalization model further.

### 5. Conclusions

In this paper, flow duration curves (FDCs) of 64 mountainous basins in eastern China were studied. It was found that the power function was the best model among the five tested functional models in the mountainous areas. Then, the two-parameters power model and 23 arbitrarily-selected basin descriptors were used to develop the regionalization models. The results show that the main controlling factors of FDCs are rainfall features and topography features. In addition, average annual rainfall (P), average elevation (H), average gradient (β), average topographic index (TI), and maximum 7d of annual rainfall

(Max7d) are determined as the best independent variables for the FDC regionalization model in the mountainous region. The FDCs have been simulated through the regionalization models, with good agreement for more than 95% of basins. We found that the developed methodology may be applied to ungauged basins since only the recorded data of the variables P, H, β, TI and Max7d are required. However, the premise of parameter transfer is that all the donors and receivers should be in a homogeneous region. In cases of application to large areas, it is necessary to determine the homogeneous regions with the hydrological regionalization method.

**Author Contributions:** Conceptualization, M.G., J.L. and Y.Y.; Methodology, S.Y., P.W. and M.G.; Software, S.Y.; Validation, S.Y.; Formal analysis, S.Y. and M.G.; Investigation, S.Y.; Resources, S.Y.; Data curation, S.Y. and M.G.; Writing – original draft, S.Y.; Writing – review & editing, J.L. and P.W.; Visualization, S.Y.; Supervision, J.L.; Project administration, J.L.; Funding acquisition, J.L. All authors have read and agreed to the published version of the manuscript.

**Funding:** This work was supported by the National Natural Science Foundation of China (NSFC) (grants, 41730750) and the Special Fund of State Key Laboratory of Hydrology-Water Resources and Hydraulic Engineering (grants, 522012242).

**Data Availability Statement:** The data that support the findings of this study are available from the corresponding author upon reasonable request.

**Acknowledgments:** The authors thank Meiyan Feng (PhD student at Hohai University) for her feedback and contributions to this manuscript.

**Conflicts of Interest:** The authors declare no conflict of interest.

## Appendix A

**Table A1.** 6 rainfall features of 64 basins.

| ID | Station | P (mm) | Max 1d (mm) | Max 3d (mm) | Max 7d (mm) | RR (mm/d) | FR (mm/d) |
|----|---------|--------|-------------|-------------|-------------|-----------|-----------|
| 1 | Ziluoshan | 760 | 65.24 | 37.31 | 20.68 | 5.26 | −5.12 |
| 2 | Zhongtang | 806 | 104.40 | 54.79 | 30.55 | 7.38 | −7.17 |
| 3 | Jizhong | 820 | 132.20 | 60.55 | 33.04 | 8.81 | −8.41 |
| 4 | Xiagushan | 764 | 85.67 | 44.26 | 25.01 | 7.00 | −6.74 |
| 5 | Gaocheng | 670 | 61.94 | 30.81 | 18.19 | 5.84 | −5.79 |
| 6 | Lixin | 930 | 100.90 | 50.07 | 25.87 | 8.78 | −8.65 |
| 7 | Guanzhai | 814 | 79.77 | 38.71 | 20.72 | 5.92 | −5.78 |
| 8 | Dapoling | 1097 | 84.07 | 41.38 | 21.39 | 6.20 | −5.87 |
| 9 | Luzhuang | 1016 | 106.70 | 53.84 | 26.48 | 7.97 | −7.44 |
| 10 | Tanjiahe | 1366 | 107.40 | 46.12 | 25.96 | 8.27 | −8.28 |
| 11 | Zhumadian | 985 | 109.40 | 53.30 | 26.29 | 8.85 | −8.35 |
| 12 | Nanlidian | 1251 | 111.90 | 55.71 | 29.71 | 9.48 | −8.74 |
| 13 | Peihe | 1346 | 104.00 | 48.61 | 28.34 | 11.27 | −10.80 |
| 14 | Baiqueyuan | 1287 | 106.40 | 49.94 | 27.59 | 9.49 | −9.09 |
| 15 | Huangnizhuang | 1332 | 108.30 | 60.22 | 32.40 | 9.62 | −9.29 |
| 16 | Qilin | 1347 | 116.70 | 60.11 | 33.13 | 10.80 | −10.13 |
| 17 | Zhangchong | 1407 | 108.70 | 58.07 | 32.25 | 9.95 | −9.53 |
| 18 | Bailianya | 1457 | 105.50 | 54.09 | 30.21 | 9.72 | −9.10 |
| 19 | Huanghewei | 1518 | 119.00 | 59.97 | 34.42 | 10.15 | −9.79 |
| 20 | Xiaotian | 1457 | 98.64 | 52.05 | 30.17 | 8.53 | −7.84 |
| 21 | Taoxi | 1215 | 76.17 | 39.15 | 21.55 | 6.65 | −5.99 |
| 22 | Liukou | 1788 | 136.60 | 69.43 | 41.23 | 13.17 | −13.18 |
| 23 | Dahekou | 1803 | 121.20 | 63.83 | 35.66 | 9.80 | −9.57 |
| 24 | Yuetan | 1778 | 110.90 | 59.23 | 36.41 | 9.87 | −9.53 |
| 25 | Shancha | 1973 | 88.73 | 48.99 | 29.69 | 10.63 | −10.27 |
| 26 | Wananba | 1798 | 87.51 | 45.14 | 24.30 | 9.60 | −9.69 |
| 27 | Sankouzhen | 1862 | 107.30 | 60.74 | 34.70 | 10.17 | −9.77 |
| 28 | Tunxi | 1770 | 104.40 | 57.38 | 34.78 | 8.77 | −8.29 |
| 29 | Yuxi | 1684 | 98.98 | 56.68 | 33.60 | 8.94 | −8.50 |

**Table A1.** *Cont.*

| ID | Station | P (mm) | Max 1d (mm) | Max 3d (mm) | Max 7d (mm) | RR (mm/d) | FR (mm/d) |
|----|---------|--------|-------------|-------------|-------------|-----------|-----------|
| 30 | Linxi | 1599 | 93.23 | 44.77 | 27.52 | 9.57 | −9.41 |
| 31 | Hulesi | 1574 | 85.89 | 46.88 | 26.48 | 8.33 | −7.92 |
| 32 | Misai | 1778 | 134.00 | 80.46 | 47.71 | 9.42 | −8.62 |
| 33 | Zhongzhou | 1788 | 94.34 | 52.50 | 30.79 | 10.61 | −10.30 |
| 34 | Yancun | 1602 | 86.07 | 55.76 | 32.44 | 8.83 | −8.30 |
| 35 | Baikuoban | 1527 | 93.20 | 50.29 | 30.63 | 9.80 | −9.61 |
| 36 | Shangzouban | 1509 | 81.18 | 43.62 | 28.67 | 10.46 | −10.40 |
| 37 | Yuankou | 1519 | 85.61 | 47.91 | 28.43 | 8.56 | −8.08 |
| 38 | Qingshandian | 1558 | 84.53 | 43.16 | 25.75 | 8.16 | −7.65 |
| 39 | Fenshui | 1582 | 89.13 | 45.35 | 28.03 | 8.27 | −7.50 |
| 40 | Shanjiao | 1499 | 70.17 | 35.72 | 23.71 | 8.87 | −8.68 |
| 41 | Laoshikan | 1687 | 93.08 | 41.70 | 22.24 | 6.56 | −5.92 |
| 42 | Qianyu | 1578 | 69.46 | 38.04 | 22.86 | 7.29 | −6.71 |
| 43 | Shangbao | 1579 | 83.24 | 44.34 | 27.01 | 8.19 | −7.79 |
| 44 | Qiaodongcun | 1631 | 89.97 | 47.31 | 27.18 | 8.41 | −7.90 |
| 45 | Hengtangcun | 1568 | 93.98 | 50.36 | 27.67 | 7.66 | −7.39 |
| 46 | Yubujie | 1466 | 96.25 | 45.18 | 25.32 | 8.97 | −8.43 |
| 47 | Shangxiantan | 1473 | 74.72 | 39.00 | 24.20 | 7.22 | −6.91 |
| 48 | Jiangwan | 1483 | 94.24 | 49.94 | 28.40 | 9.45 | −9.00 |
| 49 | Liantangkou | 1484 | 69.03 | 37.40 | 23.43 | 7.23 | −6.68 |
| 50 | Daixi | 1429 | 80.95 | 42.65 | 23.65 | 8.08 | −7.73 |
| 51 | Yiwufotang | 1477 | 67.59 | 38.28 | 23.87 | 5.52 | −4.83 |
| 52 | Dongyangyanxia | 1485 | 82.86 | 42.56 | 25.51 | 7.48 | −7.00 |
| 53 | Daitou | 1847 | 167.50 | 82.15 | 43.80 | 10.44 | −10.21 |
| 54 | Qiulu | 1707 | 147.20 | 73.22 | 37.67 | 8.72 | −8.55 |
| 55 | Caodian | 1492 | 93.49 | 47.53 | 26.82 | 8.28 | −7.93 |
| 56 | Shuangjiangxi | 1449 | 87.23 | 41.37 | 23.25 | 7.78 | −7.28 |
| 57 | Xixi | 1528 | 93.10 | 45.36 | 24.66 | 7.84 | −7.22 |
| 58 | Huangze | 1518 | 113.90 | 55.65 | 29.01 | 8.46 | −7.78 |
| 59 | Yutan | 1717 | 85.78 | 46.60 | 29.03 | 8.41 | −8.31 |
| 60 | Jianning(Xikou) | 1722 | 90.85 | 51.71 | 31.49 | 8.02 | −7.58 |
| 61 | Hongtian | 1610 | 85.31 | 45.96 | 26.61 | 7.71 | −7.46 |
| 62 | Chongan(Wuyishan) | 1918 | 103.10 | 59.27 | 36.48 | 8.34 | −8.13 |
| 63 | Fengyang | 1749 | 100.00 | 55.46 | 31.19 | 9.03 | −8.77 |
| 64 | Taipingkou | 1482 | 107.40 | 54.20 | 29.30 | 7.87 | −7.62 |

**Table A2.** 12 topographic features of 64 basins.

| ID | Station | A (km$^2$) | H (m) | β (°) | CC | CP | TI | HI | AS | RC | RF | D | NDVI |
|----|---------|-----------|-------|-------|-----|-----|-----|-----|-----|-----|-----|-----|------|
| 1 | Ziluoshan | 1800 | 822 | 19.37 | $-4.12 \times 10^{-5}$ | $-1.33 \times 10^{-4}$ | 6.37 | 0.29 | 0.47 | 0.21 | 0.34 | 1.10 | 0.90 |
| 2 | Zhongtang | 485 | 677 | 19.80 | $-6.44 \times 10^{-5}$ | $-1.18 \times 10^{-4}$ | 6.35 | 0.25 | 0.47 | 0.27 | 0.53 | 1.12 | 0.91 |
| 3 | Jizhong | 46 | 393 | 12.61 | $-1.83 \times 10^{-4}$ | $-1.33 \times 10^{-4}$ | 6.93 | 0.27 | 0.50 | 0.33 | 0.90 | 1.28 | 0.90 |
| 4 | Xiagushan | 354 | 468 | 15.15 | $-7.48 \times 10^{-5}$ | $-1.38 \times 10^{-4}$ | 6.69 | 0.25 | 0.45 | 0.26 | 0.62 | 1.12 | 0.87 |
| 5 | Gaocheng | 620 | 489 | 11.18 | $-5.26 \times 10^{-5}$ | $-5.34 \times 10^{-5}$ | 7.19 | 0.21 | 0.31 | 0.24 | 0.54 | 1.11 | 0.84 |
| 6 | Lixin | 77.8 | 172 | 6.62 | $-8.28 \times 10^{-5}$ | $-1.70 \times 10^{-4}$ | 7.71 | 0.18 | 0.26 | 0.16 | 0.46 | 1.19 | 0.83 |
| 7 | Guanzhai | 1124 | 173 | 8.29 | $-3.03 \times 10^{-5}$ | $-2.21 \times 10^{-5}$ | 7.60 | 0.16 | 0.20 | 0.26 | 0.86 | 1.09 | 0.87 |
| 8 | Dapoling | 1640 | 231 | 9.57 | $-2.86 \times 10^{-5}$ | $-1.18 \times 10^{-4}$ | 7.34 | 0.14 | 0.24 | 0.23 | 0.80 | 1.08 | 0.89 |
| 9 | Luzhuang | 396 | 212 | 8.46 | $-4.91 \times 10^{-5}$ | $-1.18 \times 10^{-4}$ | 7.49 | 0.17 | 0.27 | 0.29 | 0.68 | 1.15 | 0.88 |
| 10 | Tanjiahe | 152 | 281 | 15.36 | $-1.09 \times 10^{-5}$ | $-1.32 \times 10^{-4}$ | 6.67 | 0.25 | 0.53 | 0.29 | 0.56 | 1.19 | 0.91 |
| 11 | Zhumadian | 121 | 100 | 4.27 | $-4.52 \times 10^{-5}$ | $-8.54 \times 10^{-5}$ | 8.33 | 0.10 | 0.08 | 0.23 | 0.65 | 1.23 | 0.85 |
| 12 | Nanlidian | 1500 | 170 | 11.71 | $-3.68 \times 10^{-5}$ | $-9.62 \times 10^{-5}$ | 7.09 | 0.23 | 0.34 | 0.15 | 0.42 | 1.08 | 0.90 |
| 13 | Peihe | 17.9 | 435 | 21.69 | $-2.31 \times 10^{-5}$ | $-1.38 \times 10^{-4}$ | 6.12 | 0.50 | 0.74 | 0.31 | 0.54 | 1.28 | 0.93 |
| 14 | Baiqueyuan | 284 | 225 | 12.24 | $-6.38 \times 10^{-5}$ | $-1.15 \times 10^{-4}$ | 6.99 | 0.22 | 0.42 | 0.26 | 0.67 | 1.12 | 0.90 |
| 15 | Huangnizhuang | 805 | 487 | 18.04 | $-5.64 \times 10^{-5}$ | $-1.11 \times 10^{-4}$ | 6.48 | 0.27 | 0.40 | 0.19 | 0.58 | 1.09 | 0.93 |
| 16 | Qilin | 185 | 532 | 18.22 | $-1.63 \times 10^{-5}$ | $-1.22 \times 10^{-4}$ | 6.44 | 0.28 | 0.53 | 0.23 | 0.34 | 1.16 | 0.93 |
| 17 | Zhangchong | 493 | 671 | 19.02 | $-9.63 \times 10^{-5}$ | $-1.29 \times 10^{-4}$ | 6.37 | 0.34 | 0.45 | 0.17 | 0.28 | 1.12 | 0.93 |
| 18 | Bailianya | 745 | 667 | 19.25 | $-5.97 \times 10^{-5}$ | $-1.38 \times 10^{-4}$ | 6.31 | 0.34 | 0.48 | 0.21 | 0.51 | 1.11 | 0.93 |
| 19 | Huanghewei | 270 | 829 | 20.92 | $-9.66 \times 10^{-5}$ | $-1.38 \times 10^{-4}$ | 6.22 | 0.41 | 0.51 | 0.26 | 0.41 | 1.12 | 0.92 |

**Table A2.** *Cont.*

| ID | Station | A (km²) | H (m) | β (°) | CC | CP | TI | HI | AS | RC | RF | D | NDVI |
|---|---|---|---|---|---|---|---|---|---|---|---|---|---|
| 20 | Xiaotian | 372 | 595 | 20.89 | $-7.99 \times 10^{-5}$ | $-1.26 \times 10^{-4}$ | 6.26 | 0.27 | 0.39 | 0.21 | 0.44 | 1.12 | 0.93 |
| 21 | Taoxi | 1510 | 55 | 5.83 | $-2.07 \times 10^{-5}$ | $-7.39 \times 10^{-5}$ | 7.95 | 0.16 | 0.14 | 0.16 | 0.61 | 1.08 | 0.89 |
| 22 | Liukou | 101 | 562 | 24.37 | $-1.87 \times 10^{-4}$ | $-9.34 \times 10^{-5}$ | 6.13 | 0.28 | 0.43 | 0.26 | 0.43 | 1.14 | 0.94 |
| 23 | Dahekou | 409 | 432 | 22.58 | $-1.06 \times 10^{-4}$ | $-1.11 \times 10^{-4}$ | 6.23 | 0.32 | 0.44 | 0.23 | 0.53 | 1.12 | 0.94 |
| 24 | Yuetan | 954 | 428 | 21.22 | $-6.28 \times 10^{-5}$ | $-9.31 \times 10^{-5}$ | 6.38 | 0.22 | 0.22 | 0.21 | 0.40 | 1.09 | 0.94 |
| 25 | Shancha | 57.8 | 841 | 27.13 | $-2.01 \times 10^{-4}$ | $-1.36 \times 10^{-4}$ | 5.94 | 0.37 | 0.71 | 0.31 | 0.89 | 1.17 | 0.91 |
| 26 | Wananba | 869 | 363 | 17.96 | $-5.88 \times 10^{-5}$ | $-1.22 \times 10^{-4}$ | 6.88 | 0.20 | 0.39 | 0.23 | 0.51 | 1.09 | 0.92 |
| 27 | Sankouzhen | 259 | 604 | 23.53 | $-1.10 \times 10^{-4}$ | $-1.51 \times 10^{-4}$ | 6.17 | 0.28 | 0.61 | 0.28 | 0.53 | 1.14 | 0.93 |
| 28 | Tunxi | 2670 | 381 | 18.81 | $-4.39 \times 10^{-5}$ | $-1.05 \times 10^{-4}$ | 6.65 | 0.19 | 0.38 | 0.19 | 0.59 | 1.07 | 0.92 |
| 29 | Yuxi | 1599 | 383 | 17.99 | $-4.78 \times 10^{-5}$ | $-1.37 \times 10^{-4}$ | 6.57 | 0.19 | 0.40 | 0.18 | 0.60 | 1.09 | 0.91 |
| 30 | Linxi | 585 | 391 | 18.41 | $-7.10 \times 10^{-5}$ | $-8.65 \times 10^{-5}$ | 6.50 | 0.23 | 0.45 | 0.19 | 0.45 | 1.11 | 0.91 |
| 31 | Hulesi | 492 | 436 | 21.53 | $-7.28 \times 10^{-5}$ | $-1.58 \times 10^{-4}$ | 6.30 | 0.28 | 0.41 | 0.28 | 0.81 | 1.11 | 0.93 |
| 32 | Misai | 793 | 450 | 22.10 | $-6.59 \times 10^{-5}$ | $-1.41 \times 10^{-4}$ | 6.26 | 0.30 | 0.55 | 0.21 | 0.53 | 1.09 | 0.92 |
| 33 | Zhongzhou | 257 | 599 | 25.43 | $-9.58 \times 10^{-5}$ | $-8.94 \times 10^{-5}$ | 6.01 | 0.38 | 0.60 | 0.30 | 0.61 | 1.12 | 0.93 |
| 34 | Yancun | 180 | 546 | 24.75 | $-1.49 \times 10^{-4}$ | $-1.65 \times 10^{-4}$ | 6.11 | 0.34 | 0.61 | 0.23 | 0.48 | 1.17 | 0.91 |
| 35 | Baikuoban | 180 | 493 | 26.09 | $-1.56 \times 10^{-4}$ | $-1.57 \times 10^{-4}$ | 5.94 | 0.29 | 1.04 | 0.28 | 0.32 | 1.17 | 0.91 |
| 36 | Shangzouban | 42.9 | 381 | 23.18 | $-2.88 \times 10^{-4}$ | $-2.09 \times 10^{-4}$ | 6.15 | 0.33 | 0.64 | 0.36 | 0.66 | 1.23 | 0.92 |
| 37 | Yuankou | 687 | 270 | 18.72 | $-8.58 \times 10^{-5}$ | $-1.87 \times 10^{-4}$ | 6.52 | 0.23 | 0.43 | 0.21 | 0.50 | 1.11 | 0.91 |
| 38 | Qingshandian | 1429 | 587 | 22.95 | $-5.23 \times 10^{-5}$ | $-1.54 \times 10^{-4}$ | 6.24 | 0.32 | 0.57 | 0.22 | 0.48 | 1.10 | 0.92 |
| 39 | Fenshui | 2640 | 476 | 21.64 | $-4.20 \times 10^{-5}$ | $-1.67 \times 10^{-4}$ | 6.24 | 0.27 | 0.51 | 0.17 | 0.54 | 1.07 | 0.92 |
| 40 | Shanjiao | 189 | 658 | 26.32 | $-1.29 \times 10^{-4}$ | $-1.40 \times 10^{-4}$ | 5.93 | 0.45 | 0.62 | 0.24 | 0.26 | 1.15 | 0.94 |
| 41 | Laoshikan | 241 | 541 | 21.91 | $-8.32 \times 10^{-5}$ | $-1.53 \times 10^{-4}$ | 6.24 | 0.31 | 0.69 | 0.24 | 0.46 | 1.15 | 0.93 |
| 42 | Qianyu | 333 | 264 | 18.46 | $-9.66 \times 10^{-5}$ | $-1.92 \times 10^{-4}$ | 6.66 | 0.28 | 0.47 | 0.24 | 0.45 | 1.13 | 0.92 |
| 43 | Shangbao | 516 | 668 | 25.67 | $-9.85 \times 10^{-5}$ | $-1.31 \times 10^{-4}$ | 5.95 | 0.42 | 0.54 | 0.22 | 0.38 | 1.09 | 0.92 |
| 44 | Qiaodongcun | 233 | 437 | 19.70 | $-9.01 \times 10^{-5}$ | $-1.77 \times 10^{-4}$ | 6.37 | 0.26 | 0.69 | 0.29 | 0.50 | 1.17 | 0.92 |
| 45 | Hengtangcun | 1316 | 277 | 16.91 | $-5.27 \times 10^{-5}$ | $-1.77 \times 10^{-4}$ | 6.69 | 0.17 | 0.37 | 0.18 | 0.55 | 1.09 | 0.91 |
| 46 | Yubujie | 289 | 81 | 8.73 | $-2.72 \times 10^{-5}$ | $-1.36 \times 10^{-4}$ | 7.73 | 0.12 | 0.32 | 0.25 | 0.47 | 1.21 | 0.89 |
| 47 | Shangxiantan | 806 | 489 | 21.92 | $-5.58 \times 10^{-5}$ | $-1.70 \times 10^{-4}$ | 6.17 | 0.29 | 0.51 | 0.24 | 0.46 | 1.11 | 0.92 |
| 48 | Jiangwan | 20.9 | 306 | 20.09 | $-2.64 \times 10^{-5}$ | $-2.29 \times 10^{-4}$ | 6.20 | 0.45 | 0.56 | 0.24 | 0.29 | 1.18 | 0.94 |
| 49 | Liantangkou | 1341 | 232 | 14.24 | $-3.59 \times 10^{-5}$ | $-1.84 \times 10^{-4}$ | 6.94 | 0.21 | 0.43 | 0.26 | 0.81 | 1.09 | 0.88 |
| 50 | Daixi | 162 | 203 | 16.50 | $-1.32 \times 10^{-5}$ | $-2.31 \times 10^{-4}$ | 6.55 | 0.28 | 0.62 | 0.21 | 0.51 | 1.16 | 0.92 |
| 51 | Yiwufotang | 2341 | 285 | 15.55 | $-3.77 \times 10^{-5}$ | $-1.60 \times 10^{-4}$ | 6.84 | 0.25 | 0.46 | 0.14 | 0.56 | 1.07 | 0.88 |
| 52 | Dongyangyanxia | 830 | 305 | 16.74 | $-5.90 \times 10^{-5}$ | $-1.99 \times 10^{-4}$ | 6.66 | 0.22 | 0.42 | 0.17 | 0.49 | 1.07 | 0.89 |
| 53 | Daitou | 343 | 456 | 21.71 | $-8.10 \times 10^{-5}$ | $-1.20 \times 10^{-4}$ | 6.30 | 0.38 | 0.78 | 0.27 | 0.64 | 1.11 | 0.91 |
| 54 | Qiulu | 269 | 491 | 23.28 | $-7.91 \times 10^{-5}$ | $-1.12 \times 10^{-4}$ | 6.16 | 0.38 | 0.65 | 0.23 | 0.51 | 1.12 | 0.92 |
| 55 | Caodian | 253 | 602 | 24.33 | $-7.86 \times 10^{-5}$ | $-1.47 \times 10^{-4}$ | 6.08 | 0.37 | 0.57 | 0.29 | 0.50 | 1.13 | 0.92 |
| 56 | Shuangjiangxi | 356 | 297 | 19.72 | $-9.91 \times 10^{-5}$ | $-1.93 \times 10^{-4}$ | 6.32 | 0.37 | 0.61 | 0.21 | 0.55 | 1.12 | 0.92 |
| 57 | Xixi | 300 | 498 | 17.54 | $-9.03 \times 10^{-5}$ | $-1.76 \times 10^{-4}$ | 6.52 | 0.50 | 0.37 | 0.23 | 0.33 | 1.13 | 0.90 |
| 58 | Huangze | 542 | 310 | 17.21 | $-6.76 \times 10^{-5}$ | $-1.50 \times 10^{-4}$ | 6.56 | 0.29 | 0.58 | 0.17 | 0.36 | 1.10 | 0.91 |
| 59 | Yutan | 621 | 525 | 16.10 | $-7.64 \times 10^{-5}$ | $-1.11 \times 10^{-4}$ | 6.77 | 0.34 | 0.38 | 0.18 | 0.48 | 1.09 | 0.92 |
| 60 | Jianning (Xikou) | 1354 | 556 | 16.69 | $-4.72 \times 10^{-5}$ | $-1.06 \times 10^{-4}$ | 6.72 | 0.20 | 0.25 | 0.20 | 0.62 | 1.06 | 0.92 |
| 61 | Hongtian | 1074 | 722 | 20.39 | $-5.11 \times 10^{-5}$ | $-1.03 \times 10^{-4}$ | 6.36 | 0.38 | 0.51 | 0.24 | 0.43 | 1.09 | 0.93 |
| 62 | Chongan (Wuyishan) | 1078 | 723 | 21.62 | $-5.51 \times 10^{-5}$ | $-1.06 \times 10^{-4}$ | 6.28 | 0.27 | 0.46 | 0.21 | 0.66 | 1.10 | 0.93 |
| 63 | Fengyang | 271 | 824 | 17.88 | $-9.07 \times 10^{-5}$ | $-1.46 \times 10^{-4}$ | 6.46 | 0.29 | 0.46 | 0.19 | 0.53 | 1.14 | 0.88 |
| 64 | Taipingkou | 244 | 527 | 18.39 | $-8.80 \times 10^{-5}$ | $-1.77 \times 10^{-4}$ | 6.41 | 0.44 | 0.65 | 0.19 | 0.56 | 1.13 | 0.91 |

**Table A3.** 5 soil and geological features of 64 basins.

| ID | Station | AW | SHC (cm/s) | MBD (g/cm³) | IR (%) | FLD (km/km²) |
|---|---|---|---|---|---|---|
| 1 | Ziluoshan | 104.96 | $3.46 \times 10^{-4}$ | 1.55 | 0.84 | 0.04 |
| 2 | Zhongtang | 110.79 | $2.05 \times 10^{-4}$ | 1.55 | 0.95 | 0.06 |
| 3 | Jizhong | 102.46 | $2.54 \times 10^{-4}$ | 1.55 | 0.95 | 0.00 |
| 4 | Xiagushan | 116.62 | $3.60 \times 10^{-4}$ | 1.55 | 0.48 | 0.07 |
| 5 | Gaocheng | 103.29 | $3.46 \times 10^{-4}$ | 1.56 | 0.10 | 0.07 |
| 6 | Lixin | 118.29 | $5.65 \times 10^{-5}$ | 1.47 | 1.00 | 0.03 |
| 7 | Guanzhai | 116.62 | $8.47 \times 10^{-5}$ | 1.49 | 0.33 | 0.06 |
| 8 | Dapoling | 120.79 | $2.47 \times 10^{-4}$ | 1.53 | 0.26 | 0.12 |
| 9 | Luzhuang | 114.12 | $8.47 \times 10^{-5}$ | 1.49 | 0.51 | 0.08 |

**Table A3.** *Cont.*

| ID | Station | AW | SHC (cm/s) | MBD (g/cm$^3$) | IR (%) | FLD (km/km$^2$) |
|---|---|---|---|---|---|---|
| 10 | Tanjiahe | 102.46 | $2.12 \times 10^{-4}$ | 1.54 | 0.91 | 0.11 |
| 11 | Zhumadian | 116.62 | $4.24 \times 10^{-5}$ | 1.46 | 0.00 | 0.03 |
| 12 | Nanlidian | 118.29 | $1.91 \times 10^{-4}$ | 1.52 | 0.32 | 0.13 |
| 13 | Peihe | 114.12 | $1.77 \times 10^{-4}$ | 1.53 | 1.00 | 0.00 |
| 14 | Baiqueyuan | 125.78 | $2.61 \times 10^{-4}$ | 1.50 | 0.72 | 0.06 |
| 15 | Huangnizhuang | 119.12 | $2.47 \times 10^{-4}$ | 1.46 | 0.65 | 0.11 |
| 16 | Qilin | 122.45 | $2.26 \times 10^{-4}$ | 1.46 | 0.28 | 0.16 |
| 17 | Zhangchong | 125.78 | $2.19 \times 10^{-4}$ | 1.46 | 0.26 | 0.00 |
| 18 | Bailianya | 124.12 | $2.61 \times 10^{-4}$ | 1.43 | 0.40 | 0.04 |
| 19 | Huanghewei | 120.79 | $2.19 \times 10^{-4}$ | 1.46 | 0.48 | 0.03 |
| 20 | Xiaotian | 123.28 | $2.40 \times 10^{-4}$ | 1.46 | 0.71 | 0.07 |
| 21 | Taoxi | 121.62 | $1.98 \times 10^{-4}$ | 1.46 | 0.04 | 0.02 |
| 22 | Liukou | 114.95 | $4.24 \times 10^{-5}$ | 1.39 | 0.00 | 0.00 |
| 23 | Dahekou | 114.12 | $3.53 \times 10^{-5}$ | 1.35 | 0.02 | 0.05 |
| 24 | Yuetan | 115.79 | $4.24 \times 10^{-5}$ | 1.39 | 0.01 | 0.04 |
| 25 | Shancha | 136.61 | $1.06 \times 10^{-4}$ | 1.38 | 0.43 | 0.12 |
| 26 | Wananba | 115.79 | $9.18 \times 10^{-5}$ | 1.43 | 0.17 | 0.05 |
| 27 | Sankouzhen | 131.61 | $9.18 \times 10^{-5}$ | 1.37 | 0.26 | 0.03 |
| 28 | Tunxi | 115.79 | $7.06 \times 10^{-5}$ | 1.42 | 0.10 | 0.06 |
| 29 | Yuxi | 113.29 | $7.77 \times 10^{-5}$ | 1.44 | 0.18 | 0.07 |
| 30 | Linxi | 115.79 | $4.94 \times 10^{-5}$ | 1.39 | 0.25 | 0.11 |
| 31 | Hulesi | 117.45 | $4.94 \times 10^{-5}$ | 1.40 | 0.24 | 0.13 |
| 32 | Misai | 112.46 | $3.53 \times 10^{-5}$ | 1.40 | 0.40 | 0.09 |
| 33 | Zhongzhou | 99.96 | $2.68 \times 10^{-4}$ | 1.48 | 0.27 | 0.00 |
| 34 | Yancun | 109.12 | $7.77 \times 10^{-5}$ | 1.47 | 0.07 | 0.09 |
| 35 | Baikuoban | 118.29 | $2.12 \times 10^{-5}$ | 1.35 | 0.00 | 0.05 |
| 36 | Shangzouban | 98.29 | $9.18 \times 10^{-5}$ | 1.46 | 0.12 | 0.00 |
| 37 | Yuankou | 112.46 | $7.77 \times 10^{-5}$ | 1.44 | 0.52 | 0.07 |
| 38 | Qingshandian | 109.96 | $3.53 \times 10^{-5}$ | 1.39 | 0.38 | 0.07 |
| 39 | Fenshui | 113.29 | $2.82 \times 10^{-5}$ | 1.38 | 0.27 | 0.06 |
| 40 | Shanjiao | 114.95 | $3.53 \times 10^{-5}$ | 1.38 | 0.95 | 0.00 |
| 41 | Laoshikan | 114.95 | $4.24 \times 10^{-5}$ | 1.39 | 0.63 | 0.10 |
| 42 | Qianyu | 110.79 | $2.12 \times 10^{-5}$ | 1.36 | 0.25 | 0.02 |
| 43 | Shangbao | 116.62 | $2.82 \times 10^{-5}$ | 1.36 | 0.78 | 0.09 |
| 44 | Qiaodongcun | 116.62 | $3.53 \times 10^{-5}$ | 1.36 | 0.50 | 0.00 |
| 45 | Hengtangcun | 114.95 | $6.35 \times 10^{-5}$ | 1.42 | 0.39 | 0.06 |
| 46 | Yubujie | 119.12 | $2.19 \times 10^{-4}$ | 1.47 | 0.09 | 0.00 |
| 47 | Shangxiantan | 117.45 | $4.24 \times 10^{-5}$ | 1.39 | 0.75 | 0.13 |
| 48 | Jiangwan | 106.62 | $1.13 \times 10^{-4}$ | 1.48 | 1.00 | 0.00 |
| 49 | Liantangkou | 108.29 | $1.55 \times 10^{-4}$ | 1.47 | 0.58 | 0.06 |
| 50 | Daixi | 111.62 | $7.06 \times 10^{-5}$ | 1.48 | 0.82 | 0.02 |
| 51 | Yiwufotang | 111.62 | $9.18 \times 10^{-5}$ | 1.45 | 0.63 | 0.09 |
| 52 | Dongyangyanxia | 112.46 | $7.06 \times 10^{-5}$ | 1.44 | 0.73 | 0.13 |
| 53 | Daitou | 118.29 | $3.53 \times 10^{-5}$ | 1.41 | 0.26 | 0.00 |
| 54 | Qiulu | 114.95 | $2.82 \times 10^{-5}$ | 1.37 | 1.00 | 0.00 |
| 55 | Caodian | 114.95 | $3.53 \times 10^{-5}$ | 1.43 | 0.77 | 0.09 |
| 56 | Shuangjiangxi | 107.46 | $5.65 \times 10^{-5}$ | 1.43 | 0.86 | 0.15 |
| 57 | Xixi | 120.79 | $7.06 \times 10^{-5}$ | 1.42 | 0.94 | 0.00 |
| 58 | Huangze | 114.95 | $4.94 \times 10^{-5}$ | 1.40 | 0.86 | 0.04 |
| 59 | Yutan | 119.12 | $4.24 \times 10^{-5}$ | 1.40 | 0.44 | 0.13 |
| 60 | Jianning(Xikou) | 120.79 | $4.94 \times 10^{-5}$ | 1.41 | 0.34 | 0.11 |
| 61 | Hongtian | 114.95 | $4.24 \times 10^{-5}$ | 1.42 | 0.66 | 0.06 |
| 62 | Chongan(Wuyishan) | 117.45 | $5.65 \times 10^{-5}$ | 1.41 | 0.75 | 0.04 |
| 63 | Fengyang | 119.12 | $8.47 \times 10^{-5}$ | 1.42 | 0.90 | 0.02 |
| 64 | Taipingkou | 119.12 | $3.53 \times 10^{-5}$ | 1.39 | 0.93 | 0.08 |

**Table A4.** The *NSE* and ε values of simulated FDCs to the observed FDCs for each of the 64 basins.

| ID | Station | *NSE* | ε (%) | ID | Station | *NSE* | ε (%) |
|---|---|---|---|---|---|---|---|
| 1 | Ziluoshan | 0.26 | 18.16 | 33 | Zhongzhou | 0.97 | 16.16 |
| 2 | Zhongtang | 0.77 | 16.42 | 34 | Yancun | 0.93 | 14.41 |
| 3 | Jizhong | 0.99 | 12.07 | 35 | Baikuoban | 0.99 | 13.72 |
| 4 | Xiagushan | 0.74 | 15.20 | 36 | Shangzouban | 0.96 | 12.87 |
| 5 | Gaocheng | 0.34 | 100.50 | 37 | Yuankou | 0.98 | 15.67 |
| 6 | Lixin | 0.81 | 13.60 | 38 | Qingshandian | 0.98 | 13.78 |
| 7 | Guanzhai | 0.91 | 11.90 | 39 | Fenshui | 0.98 | 14.41 |
| 8 | Dapoling | 1.00 | 13.92 | 40 | Shanjiao | 0.93 | 13.13 |
| 9 | Luzhuang | 0.96 | 16.95 | 41 | Laoshikan | 0.62 | 15.80 |
| 10 | Tanjiahe | 0.93 | 18.01 | 42 | Qianyu | 0.95 | 18.44 |
| 11 | Zhumadian | 0.99 | 18.63 | 43 | Shangbao | 0.99 | 14.30 |
| 12 | Nanlidian | 0.99 | 15.30 | 44 | Qiaodongcun | 0.96 | 15.76 |
| 13 | Peihe | 0.97 | 18.09 | 45 | Hengtangcun | 0.83 | 11.35 |
| 14 | Baiqueyuan | 0.94 | 14.53 | 46 | Yubujie | 0.96 | 15.58 |
| 15 | Huangnizhuang | 0.99 | 14.30 | 47 | Shangxiantan | 0.96 | 16.10 |
| 16 | Qilin | 0.94 | 11.70 | 48 | Jiangwan | 0.98 | 17.86 |
| 17 | Zhangchong | 1.00 | 11.39 | 49 | Liantangkou | 0.98 | 15.95 |
| 18 | Bailianya | 0.98 | 11.44 | 50 | Daixi | 0.98 | 17.79 |
| 19 | Huanghewei | 0.98 | 8.24 | 51 | Yiwufotang | 0.98 | 14.52 |
| 20 | Xiaotian | 0.96 | 13.53 | 52 | Dongyangyanxia | 0.98 | 16.79 |
| 21 | Taoxi | 0.92 | 30.82 | 53 | Daitou | 0.99 | 12.74 |
| 22 | Liukou | 0.96 | 14.30 | 54 | Qiulu | 0.99 | 11.69 |
| 23 | Dahekou | 0.99 | 15.79 | 55 | Caodian | 0.97 | 16.19 |
| 24 | Yuetan | 0.99 | 14.31 | 56 | Shuangjiangxi | 0.98 | 18.03 |
| 25 | Shancha | 0.95 | 16.72 | 57 | Xixi | 0.97 | 15.45 |
| 26 | Wananba | 0.95 | 16.37 | 58 | Huangze | 0.91 | 15.74 |
| 27 | Sankouzhen | 0.98 | 13.73 | 59 | Yutan | 0.98 | 10.14 |
| 28 | Tunxi | 0.98 | 14.97 | 60 | Jianning | 0.97 | 7.42 |
| 29 | Yuxi | 0.99 | 15.33 | 61 | Hongtian | 0.69 | 9.62 |
| 30 | Linxi | 0.97 | 15.41 | 62 | Chongan | 0.99 | 11.77 |
| 31 | Hulesi | 0.99 | 15.25 | 63 | Fengyang | 0.97 | 9.53 |
| 32 | Misai | 0.99 | 15.06 | 64 | Taipingkou | 0.98 | 9.46 |

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
