# Peer review of "Regionalizing Streamflow Regime Function through Integrations of Geographical Controls in Mountainous Basins"

_water, doi:10.3390/w15020280_

Round 1
Reviewer 1 Report
Authors explored the integrations of geographical controls on the regional streamflow regime function, so as to develop accurate regional FDCs to solve the problem of runoff prediction in the ungauged mountainous basins of the eastern China. It is a good paper with sound research content, structure, illustration and discussion. There are some small problems should be clarified.
Comments and suggestions:
1) Introduction. “1.1. Objectives”. I suggest author should combine this part into introduction.
2) Line 160, “13 pairs of (QD, D)” should be emphasized with “13 flow percentiles, including Q1, Q5, Q10, Q20, Q30, Q40, Q50, Q60, Q70, Q80, Q90, Q95, Q99”.
3) Table 1 is “Description of the 23 basin descriptors”. I suggest authors should give one detailed table showing the concrete value of these descriptors in 23 basin.
4) Figure 4 is very important in the manuscript. Authors should give more description. Why there is no value (-) in some percentile flows? “average gradient (β)” has negative and positive coefficients for different percentile flows, and why?
Reviewer 2 Report
Although the authors emphasize the importance of homogeneity for regionalization (see lines 397-398), they do not propose any ideas about criteria for the identification of homogeneous regions, particularly in combination with the hydrological regionalization method. Could not for instance the deviation between the values of some descriptors of the receiver basin and the corresponding values of the donor basin (as it was found for the basins Ziluoshan and Gaocheng, see lines 391-392), be an indication of inhomogeneity?
